


# Multiple solvent signal presaturation in $^{13}$C NMR

Marine Canton[1,2], Richard Roe[2], Stéphane Poigny[2], Jean–Hugues Renault[1], and Jean–Marc Nuzillard[1]

[1]Université de Reims Champagne Ardenne, CNRS, ICMR UMR 7312, 51097 Reims, France
[2]Laboratoires Pierre Fabre Dermocosmétique, 3 Avenue Hubert Curien, BP 13562, 31035 Toulouse Cedex, France

**Correspondence:** M. Canton (marine.canton@univ-reims.fr), J.–M. Nuzillard (jm.nuzillard@univ-reims.fr)

**Abstract.** The analysis by proton-decoupled carbon-13 nuclear magnetic resonance spectroscopy of samples dissolved in solvents presenting strong multiple resonances can be facilitated by the suppression of these resonances by multi–site presaturation. The advantage drawn from this operation is the elimination of the possible artifacts that arise from the solvent signals in non–optimized decoupling conditions. Solvent presaturation was implemented on glycerol, 1,2–propanediol, 1,3–propanediol, 1,2–butanediol, 1,3–butanediol with at least 94 % on–resonance efficiency and a bandwidth of less than 50 Hz measured at 50 % signal intensity decrease. The experimental measurement of the signal suppression bandwidth leads to unexpected selectivity profiles for frequency close resonances. Computer resolution of the Bloch equations during multi–site presaturation provide an insight into the origin of the observed profile perturbations.

## 1 Introduction

Nuclear magnetic resonance (NMR) is the only spectroscopic method used for the structural elucidation of organic molecules that produces information at the atomic level. Liquid state NMR of proteins strongly relies on the observation of the amide NH proton resonances and is therefore carried out in a solvent mainly composed of light water. The concentration of hydrogen in protein NMR samples ($110 \, \mathrm{mol \, L^{-1}}$) compared to the one of the protein itself ($1 \, \mathrm{mmol \, L^{-1}}$ or less, (Zheng and Price, 2010)) forced NMR spectroscopists to create efficient water signal suppression techniques (Lee et al., 2017; Chen et al., 2017; Duarte et al., 2013; Gouilleux et al., 2017). Without them, the water signal would cover a wide band of signals of high structural importance and would also hamper the accurate operation of analog to digital signal conversion devices (Mo and Raftery, 2008) resulting in detection sensitivity reduction. Small molecule NMR also benefits from solvent signal suppression techniques when hyphenated to liquid chromatography in the study of fluids of biological (plasma, urine, ...) or food (fruit juices, alcoholic beverages, ...) interest (Friedbolin, 2011; Kew et al., 2017).

A high signal rejection ratio, a low pertubation of the baseline, and a narrow signal attenuation frequency window define a high quality of a solvent signal suppression technique (Zheng and Price, 2010). A narrow suppression window ensures that the intensities of resonances close to the one of the solvent will be preserved at best. Solvent resonance presaturation is the oldest





of these techniques and consists in the application during the relaxation delay of a low power radiofrequency (RF) field on
resonance with the solvent signal (Hoult, 1976) (Ross et al., 2007).

    Multiple solvent signal suppression is a necessity in LC-NMR and for the study of alcoholic beverages by [1]H NMR. In the latter case, the eight signals produced by water and ethanol can be efficiently attenuated (Monakhova et al., 2011). However, the presence of solvents is not a problem in [13]C NMR spectroscopy since their resonance lines are very sharp, relatively to the width of the observation frequency window, and are not likely to overlap those of interest. The context of the present
study is the characterization by [13]C NMR of compounds within natural extracts (Hubert et al., 2014) (Tsujimoto et al., 2018) (Bakiri et al., 2017). Plant extracts may be conditioned as dry products or as solutions in diverse solvents, possibly prepared from renewable resources and for which evaporation to dryness may be not feasible or not compatible with the chemical integrity of the solutes. Indeed, alcohols like glycerol, propanediols, butanediols and pentanediols are employed for such applications (Chemat et al., 2019) (Shehata et al., 2015). Their boiling points range from 188 °C to 290 °C under atmospheric
pressure. The characterization of the solutes by [13]C NMR spectroscopy can be carried out on extracts or on fractions obtained by chromatographic methods. The fractions of interest may also contain an important amount of these high boiling point solvents.

    NMR data acquisition of series of samples is often carried out in automation mode with standard acquisition parameters. An accurate calibration of pulses on the [1]H RF channel is necessary to record [13]C–{[1]H} spectra in proper decoupling conditions.
Miscalibration may cause decoupling artifacts around the intense solvent signals, at a point their intensity is comparable to the one of the signals of interest (Blechta and Schraml, 2015).

    Analytically misleading decoupling artifacts were observed during the analysis by [13]C NMR of chromatographic fractions containing glycerol. Their elimination through the reduction of their parent glycerol signals was achieved by multi–site presaturation, using multiple modulation of the RF field (Patt, 1992).

The assessment of the method included the determination of the frequency profile of signal attenuation around the presaturation frequencies. Samples that contain 1,2–propanediol show [13]C NMR spectra with two close resonance lines, a few Hz apart from each other, depending on concentration. The corresponding saturation profile showed unexpected features that incited us to investigate in detail the underlying spin dynamics by numerical simulation. The apparent interference effect between saturation pulses recalled the one observed for two closely frequency-shifted BURP pulses, as reported in the article entitled
"Close encounters between soft pulses" (Kupče and Freeman, 1995). In this article, Ē. Kupče and R. Freeman demonstrated that when the difference between the two frequency shifts has the same order of magnitude as the selective pulse operation bandwidth, then the resulting operation frequency profile presents a cahotic aspect.

    The first part of the following section deals with simple theoretical aspects of presaturation. Experimental results include the study of a sucrose sample diluted in glycerol and show that presaturation is effective for decoupling artefact removal and
the handling of other solvents that present up to four resonances such as 1,2–propanediol, 1,3–propanediol, 1,2–butanediol and 1,3–butanediol.





## 2 Theory

Resonance saturation in NMR occurs when an RF field is continuously applied at a frequency equal to the resonance frequency
of a nucleus. The magnetization dynamics of a collection of many identical isolated spins that constitutes a macroscopic sample
is governed by the Bloch equations (Bloch, 1946). The components $M_x$, $M_y$, and $M_z$ of the macroscopic magnetization $\boldsymbol{M}$,
when observed in the rotating frame of reference, evolve as follows,

$$
\begin{aligned}
\frac{dM_x}{dt} &= \Omega_0 M_y - \Omega_{1y} M_z - R_2 M_x \\
\frac{dM_y}{dt} &= \Omega_{1x} M_z - \Omega_0 M_x - R_2 M_y \\
\frac{dM_z}{dt} &= \Omega_{1y} M_x - \Omega_{1x} M_y - R_1 (M_z - M_z^{\mathrm{eq}})
\end{aligned}
\tag{1}
$$

in which $\Omega_0$ is the precession angular frequency of the nuclei, $\Omega_1$ is the norm of the nutation vector expressed as an angular
frequency, and $(\Omega_{1x}, \Omega_{1y})$, the components of the latter on the $X$ and $Y$ axis of the rotating frame. Nuclear spin relaxation is
phenomenologically described by the two rate constants $R_1$ and $R_2$ defined as the reciprocals of the longitudinal and transverse
relaxation times $T_1$ and $T_2$, respectively. $M_z^{\mathrm{eq}}$ denotes the value of the sample equilibrium nuclear magnetization and intervenes
in the description of the longitudinal relaxation. In the case $\Omega_0 = 0$ of an on-resonance constant intensity applied RF field, the
components of the magnetization vector tend toward a stationary limit for which

$$
M_z^{\mathrm{stat}} = \frac{M_z^{\mathrm{eq}}}{1 + \Omega_1^2 T_1 T_2}
\tag{2}
$$

If $\Omega_1^2 T_1 T_2 \gg 1$, then the stationary magnetization is much lower than the one of equilibrium, corresponding to an equalization
of spin state populations induced by the RF field, as expected from saturation.

Solvent signal suppression in NMR spectroscopy can be obtained by selective saturation of one or more solvent signals
during the relaxation delay. This technique is named presaturation because it precedes the non–selective excitation of the
sample resonances. Presaturation at a single site is easily achieved by continuous wave RF irradiation. Multi–site presaturation
relies on multiple–frequency–shifted laminar pulses, a particular species of shaped pulse (Patt, 1992). Such a shaped pulse
serves as presaturation module of duration $T$ and is applied repetitively to the sample so that the overall RF irradiation time is
equal to the desired relaxation delay. A presaturation module is constituted by $N$ elementary pulses, named slices hereafter, of
duration $\delta t$ so that $T = N \delta t$. The creation of a module requires the definition of $T$, $N$, of the number $n$ of presaturation sites,
and of the list of the frequency offsets $\Omega_k^{\mathrm{sat}}$ associated to each site. The values of $\Omega_{1x}$ and $\Omega_{1y}$ are obtained from

$$
(\Omega_{1x} + i\Omega_{1y})(t_j) = \frac{\Omega_1}{n} \sum_{k=0}^{n-1} \exp(i\Omega_k^{\mathrm{sat}} \cdot j\delta t) \quad \text{for} \quad 0 \leq j < N
\tag{3}
$$

which states that RF field intensities are equally distributed among the $n$ sites and phases arbitrarily are set to zero at $t = 0$.

The $\Omega_k^{\mathrm{sat}}$ values are calculated relatively to an auxiliary carrier frequency determined as the average of the highest and the
lowest offsets of the signals to presaturate. The emission of the presaturation pulse has to take into account the difference
between the auxiliary frequency and the actual transmitter frequency, the so–called shaped pulse offset, as described in Fig.



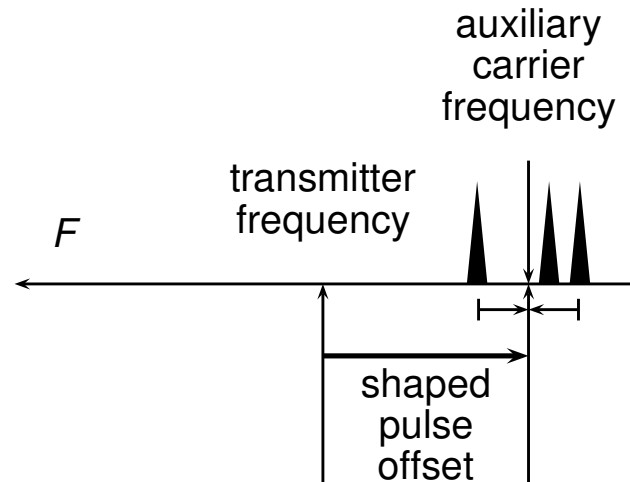

**Figure 1.** Definition of the offset of the presaturation shaped pulses in a schematic spectrum. The narrow black triangles represent the solvent peaks. The auxiliary carrier frequency is defined as the mean of the highest and lowest solvent resonance frequencies. The offset of the presaturation shaped pulse is the difference between the auxiliary carrier frequency and the actual transmitter frequency.

1. The value of $\delta t$ is chosen so that the highest precession angle $|\Omega_k^{\mathrm{sat}}|\delta t$ for the highest $|\Omega_k^{\mathrm{sat}}|$ during that time must be kept below a small threshold value in the order of $\pi/15$. The value of $N$ should be as high as possible and depends on the memory size available for shaped pulses in the pulse program sequencer. $N = 50,000$ was used throughout the present study. The $N\delta t$

product determines the shaped pulse duration $T$. Alternatively, $T$ may be chosen so that the highest precession angle during $\delta t$ falls under the predefined threshold for the retained value of $N$.

     The simulation of a set of saturation profiles like the one in Fig. 2 requires first the creation of a table of $N$ values of $\Omega_{1x}$ and of $\Omega_{1y}$ according to Eqn. (3). Nucleus resonance offset frequencies $\Omega_0/2\pi$ are then repetitively selected for presaturation effect calculation from a set of linearly spaced values comprised between a minimum and a maximum. Starting from a magnetization

vector in its equilibrium position, the action it undergoes from the series of presaturation modules is evaluated. The offset frequency and final amount of longitudinal magnetization $M_z$ are printed in a computer file so that a graph of $M_z(\Omega_0/2\pi)$ can be drawn for the chosen set of $\Omega_0$ values. The action of a presaturation module is determined by the action of the series of its constituting slices. The action of each shaped pulse slice should be calculated by resolution of the Bloch equation system (Eqn. 1) over duration $\delta t$, even though a different method was followed, as explained hereafter.

Exact solutions of the Bloch equations have been reported but bear some degree of complexity (Canet et al., 1994; Madhu and Kumar, 1995). They take account of magnetization precession, nutation and relaxation processes simultaneously. The approach followed here makes use of an easy to implement approximate solution. It relies on the observation that magnetization evolution induced by relaxation alone is slow compared to the one induced by simultaneous precession and nutation. The evolution of $\boldsymbol{M}$ solely under precession and nutation resumes to a rotation at angular frequency $\Omega^{\mathrm{eff}}$, the norm of vector $\boldsymbol{\Omega}^{\mathbf{eff}}(\Omega_{1x}, \Omega_{1y}, \Omega_0)$

when reported in the rotating frame of reference. The rotation axis is defined by the unitary vector $\boldsymbol{u} = \boldsymbol{\Omega}^{\mathbf{eff}}/\Omega^{\mathrm{eff}}$. For practical





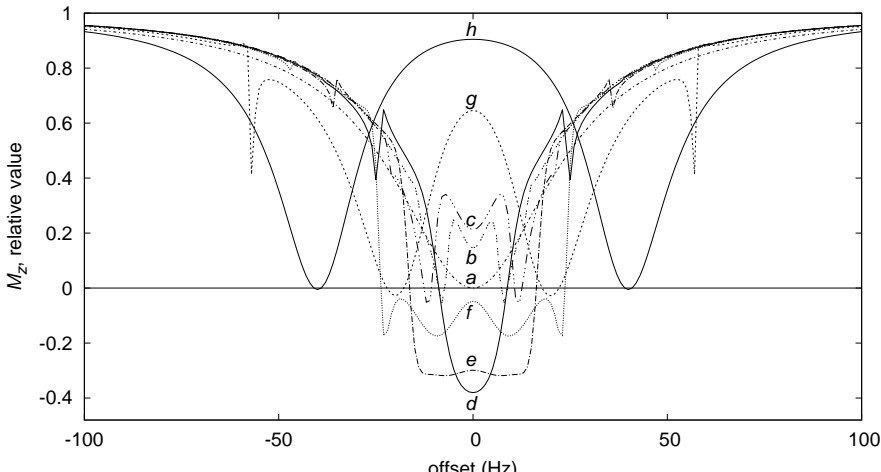

**Figure 2.** Saturation profiles for presaturation at two sites , at $\pm\Omega^{\mathrm{sat}}/2\pi$ for $\Omega^{\mathrm{sat}}/2\pi = 0, 2, 4, 6, 8\ 10, 20, 40\,\mathrm{Hz}$, corresponding to traces $a$ to $h$. The relaxation times $T_1$ and $T_2$ are both equal to 0.5 s. The relaxation delay lasts 5 s during which 10 presaturation modules of 0.5 s each are applied. Each module is made of 50000 slices, for which $\Omega_{1x}$ and $\Omega_{1y}$ values are calculated with $\Omega_1/2\pi = 50\,\mathrm{Hz}$.

calculations, one needs to express the elements of the rotation matrix $\mathbf{R}_{\boldsymbol{u},\theta}$ in which $\theta = \Omega^{\mathrm{eff}}\delta t$ and $\boldsymbol{u}(u_x, u_y, u_z)$.

$$\mathbf{R}_{\boldsymbol{u},\theta} = \cos\theta \begin{pmatrix} 1 & 0 & 0 \\ 0 & 1 & 0 \\ 0 & 0 & 1 \end{pmatrix} + (1-\cos\theta) \begin{pmatrix} u_x^2 & u_x u_y & u_x u_z \\ u_x u_y & u_y^2 & u_y u_z \\ u_x u_z & u_y u_z & u_z^2 \end{pmatrix} + \sin\theta \begin{pmatrix} 0 & -u_z & u_y \\ u_z & 0 & -u_x \\ -u_y & u_x & 0 \end{pmatrix} \tag{4}$$

Relaxation alone is taken into account by the following transformation of $\boldsymbol{M}$.

$$(M_x, M_y, M_z) \longrightarrow (M_x e^{-R_2 \delta t}, M_y e^{-R_2 \delta t}, M_z^{\mathrm{eq}} + (M_z - M_z^{\mathrm{eq}})e^{-R_1 \delta t}) \tag{5}$$

The evolution of $\boldsymbol{M}$ during a time slice of duration $\delta t$ is simply calculated by the successive application of rotation and relaxation transformations. The approximation that consists in alternating rotation and relaxation instead of considering them simultaneously improves when $\delta t$ tends to zero. A given $\delta t$ time interval can be divided in two (or more) parts and the replacement of rotation($\delta t$)–relaxation($\delta t$) by two consecutive rotation($\delta t/2$)–relaxation($\delta t/2$) calculations provides a way to evaluate the error induced by the proposed calculation method.

An identical approach to Bloch equations resolution was used for the optimization of band–selective uniform response pulses (BURP) in the presence of relaxation, leading to the design of pulses with silhouette largely unaffected by relaxation processes (SLURP), for which the underlying calculation details were not reported (Nuzillard and Freeman, 1994). The action of relaxation on frequency–domain profiles of BURP pulses were recalculated using exact solutions of the Bloch equations and the results were visually identical to those derived from the approximate treatment (Canet et al., 1994).



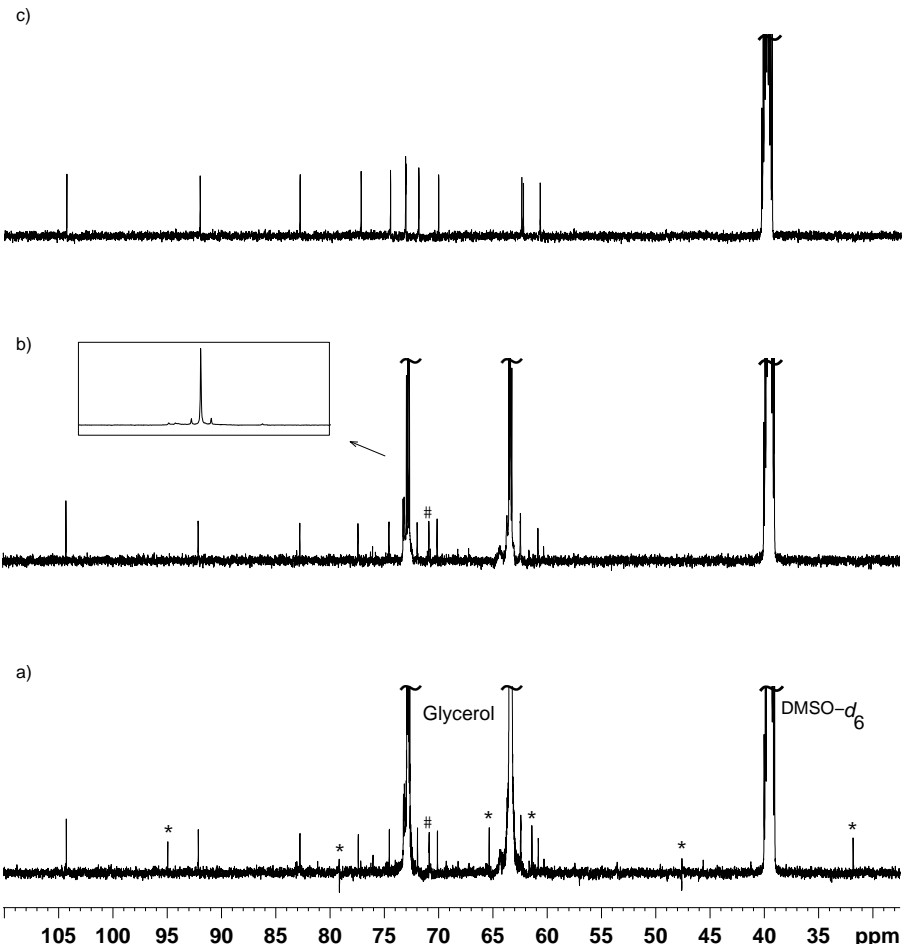

**Figure 3.** a) $^{13}$C NMR spectrum of D(+)–sucrose (24 mM) and glycerol (3.6 M) in DMSO–$d_6$. The "*" sign indicates decoupling artifacts. The "#" sign indicates a signal from a minor compound contained in bio–sourced glycerol; b) Analysis of the same sample as in a) but with multiple presaturation of glycerol signals. The framed insert shows a detailed view of the peak at $\delta$ 73.1 and of its $^{13}$C satellites, drawn without vertical truncation. c) $^{13}$C NMR spectrum of D(+)–sucrose (24 mM) in DMSO–$d_6$. All acquisitions required the recording of 128 scans preceded by 8 dummy scans.

## 3   Results

The unwanted effect on $^{13}$C NMR spectra of the presence of glycerol in high concentration was reproduced by the analysis of a solution of sucrose (29 mM) in DMSO–$d_6$ to which glycerol (3.62 M) was added. This sample constitutes a good approximation of a real case, as industrially prepared plant extracts are often delivered as solutions in high boiling point solvents like glycerol, at metabolite concentrations close to or lower than that of sucrose in our model preparation.





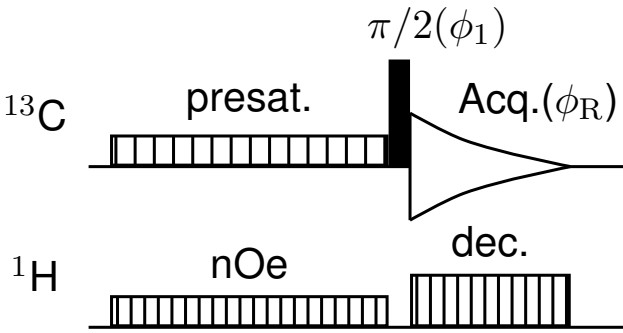

**Figure 4.** Pulse sequence for the recording of $^{13}$C–{$^1$H} spectra with sensitivity enhancement by nuclear Overhauser effect magnetization transfer from $^1$H nuclei and with presaturation of solvent resonances. The minimal phase program is $\phi_1 = \phi_R = (0, \pi)$.

Fig. 3$a$ presents the $^{13}$C NMR spectrum of sample sucrose in glycerol and its comparison with the $^{13}$C NMR spectrum of sucrose alone in DMSO–$d_6$. The spectrum in Fig. 3$c$ shows the residual signal of DMSO–$d_6$ and the twelve peaks from sucrose, two of them at $\delta\,73.13$ and $\delta\,73.15$ being not well resolved. The $^{13}$C NMR spectrum of sucrose in glycerol contains supplementary peaks, the two intense ones of glycerol apart. Glycerol clearly introduced unexpected signals in the spectra, some with abnormal phases, but others that may be considered genuine, thus creating confusion in the analysis of unknown samples.

Glycerol also introduces peaks that arise from production side–products present at very low but detectable concentrations. A possible origin of the artifact signals was first searched in a possible saturation of the spectrometer receiver or an intermodulation related problem; changing the receiver gain did not influence their position and phase, so that this hypothesis was not further considered (Marshall and Verdun, 1990). Receiver gain was set to its maximum value in all following experiments.

Broadband heteronuclear decoupling constitutes another source of artifacts in $^{13}$C–{$^1$H} NMR spectra. A proper adjustment

of power in the $^1$H channel is required for the recording of an optimal, artifact-free $^{13}$C spectrum with WALTZ-16 composite pulse decoupling (Shaka et al., 1983). Slight changes in decoupling RF power resulted in changes of position and phase of artifacts. The strongest signals being by far those of glycerol, their intensity reduction brought the decoupling artifact intensity below the noise level as shown by Fig. 3$b$. Obviously, a better calibration of the RF pulses in the decoupler channel would also reduce, if not eliminate, the decoupling artifacts. The recording of series of samples in automation mode with a sample changer

does not favor the calibration of decoupler RF pulse on a sample–to–sample basis, so that the study of strong signal reduction was undertaken.

Glycerol signal reduction in $^{13}$C NMR was achieved by presaturation. As observed in Fig. 3$b$, reducing the intensity of solvent signals by double presaturation removed decoupling artifacts and the observed signals only arose from the compounds present in the sample. This procedure was carried out on more than 30 samples of natural extracts diluted in glycerol.

The characterization of glycerol signal presaturation was further undertaken by means of a sample made only of glycerol in DMSO–$d_6$. The study relied on the pulse sequence in Fig. 4, which is a straighforward adaptation of *zgpg* from the TopSpin library, in which presaturation is implemented as the repeated emission of an RF shaped pulse. The minimal two–step phase



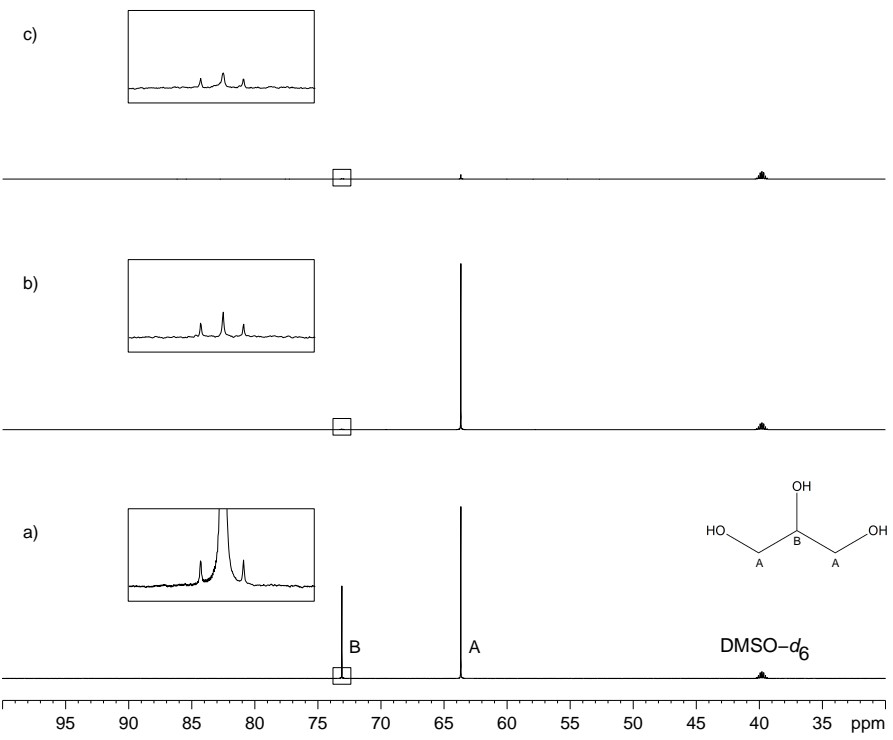

**Figure 5.** Presaturation of glycerol in DMSO–$d_6$. a) $^{13}$C NMR spectrum of glycerol. Signals from sites A and B are located at $\delta_A$ 63.7 and $\delta_B$ 73.1. b) Effect of single presaturation at site B. c) Effect simultaneous presaturation at sites A and B. The presaturation module last $T$ = 1 s and are applied with a maximal intensity $\Omega_1/2\pi$ of 11.7 Hz All acquisitions required the recording of 8 scans preceded by 4 dummy scans.

program ensures that peaks are all identically phased and that their height is proportional to the amount of longitudinal magnetization present at the end of the presaturation period. Glycerol, $C_3H_8O_3$, produces only two $^{13}$C NMR signals by symmetry, located at $\delta_A$ 63.7 and $\delta_B$ 73.1. Presaturation by continuous wave at a single site, A or B, in both cases resulted in a 99 % signal intensity reduction while simultaneous presaturation at sites A and B caused an attenuation better than than 97 %, as shown in Fig. 5.

Experimental saturation profiles were measured in order to evaluate the width of the frequency band concerned by signal attenuation. For this purpose, the frequency offset of the presaturation pulse was varied in 1 Hz steps around the value that corresponds to the on–resonance RF field application. The presaturation bandwidth is defined by the interval of frequency offsets in which signal intensity is reduced at least by 50 %. The profile of the signal from position A in glycerol presented a bell shape whose full width at half height was 15 Hz for $\Omega_1/2\pi$ = 11.7 Hz, that represents a bandwidth of 0.1 ppm at 151 MHz

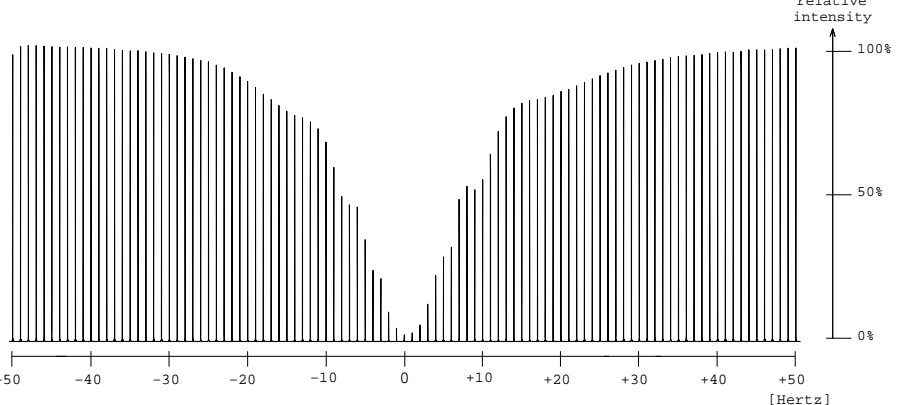

**Figure 6.** Measurement of the frequency interval width for which presaturation causes a decrease of at least 50 % of the signal A intensity ($\delta$ 63.7) by changing the auxiliary frequency in 1 Hz steps from -50 Hz to +50 Hz.

(Fig. 6). A similar width, 0.09 ppm, was measured for the presaturation at the site B. The profiles are those expected for a multi–site presaturation of two very largely separated resonances, such as those of glycerol, with a difference in peak position

of 9.43 ppm (or 1424 Hz). Such narrow zones of signal attenuation are compatible with the practical identification of the dissolved compounds.

The power of presaturation RF pulses influences the on–resonance residual longitudinal magnetization and therefore the intensity of the residual signal. This power must be low enough to keep the presaturation band sufficiently narrow and high enough to achieve a useful signal suppression. Five experiments (not shown) were carried out by reducing the power of RF

pulse intensity from 58.7 Hz to 5.9 Hz. The intensity of the two residual signals were similar: signal attenuation was always at least 95 %. Based on this result, an intensity of 11.7 Hz was retained for presaturation pulses in all subsequent spectra recordings.

Multiple site presaturation was extended to other heavy solvents used as natural product extractants: 1,2–propanediol, 1,3–propanediol, 1,2–butanediol and 1,3–butanediol. For all but 1,2–propanediol, presaturation reduced solvent signal intensity

by at least 94 %. Presaturation was also carried out on samples containing sucrose and each of the heavy solvents mentioned here above. The spectra recorded with and without presaturation as well as the corresponding raw NMR data are available for download. As expected, presaturation has resulted in a strong decrease of targeted signals and the removal of decoupling artifacts. Table 1 summarizes the results obtained for each heavy solvent, concerning signal attenuation and signal attenuation bandwidth.

The elimination of the $^{13}$C resonances of 1,2–propanediol led to an unexpected presaturation profile in the region of two oxygen–bearing carbons, due to their very close chemical shift values, 67.8 ppm and 67.9 ppm, as shown in Fig. 7. The profile showed puzzling irregular features that motivated the undertaking of a numerical simulation work. In this case $\delta\Omega^{\text{sat}}/2\pi =$ 10 Hz. The simulated profile in Fig. 2$f$, corresponding to a 10 Hz offset, has similarities with the experimental one as shown



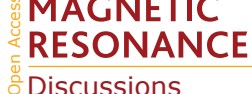

**Table 1.** Presaturation characteristics obtained on the selected heavy solvents. The '*' sign indicates a perturbation of the presaturation profile due to frequency–close resonances.

| Matrix | Chemical shift (ppm) | Attenuation (%) | Bandwidth (Hz) |
|---|---|---|---|
| Glycerol | 63.66 | 97 | 15 |
|  | 73.09 | 97 | 14 |
| 1,2–Propanediol | 20.36 | 95 | 20 |
|  | 67.83 | 95 | 42* |
|  | 67.89 | 95 | 46* |
| 1,3–Propanediol | 36.24 | 94 | 36 |
|  | 58.72 | 99 | 23 |
| 1,2–Butanediol | 10.47 | 98 | 5 |
|  | 26.68 | 99 | 5 |
|  | 66.13 | 99 | 11 |
|  | 73.13 | 99 | 6 |
| 1,3–Butanediol | 24.32 | 98 | 10 |
|  | 42.40 | 98 | 10 |
|  | 58.86 | 98 | 11 |
|  | 64.20 | 96 | 12 |

in the Fig. 7 zoom frame. Indeed, a wavy effect is also observed at ± 20 Hz offset around the resonance. This phenomenon
generates a bandwidth for the two close signals of 1,2–propanediol ($\delta$ 63.8) higher than the one for the isolated signal ($\delta$ 20.4),
respectively 46 Hz and 20 Hz. However, since 46 Hz corresponds to 0.3 ppm on our spectrometer, this result is still acceptable.

  Solvent signal suppression was automated for the five studied solvents by means of computer scripts written in C language.
The creation of the shaped pulse of the presaturation module was carried out by recording first a $^{13}C–\{^{1}H\}$ spectrum with the
*zgpg* pulse sequence, noting the solvent resonance frequencies by spectrum peak picking, calculating the $\Omega_k^{\mathrm{sat}}/2\pi$ frequencies
and the shaped pulse offset, and generating the corresponding RF waveform definition file.





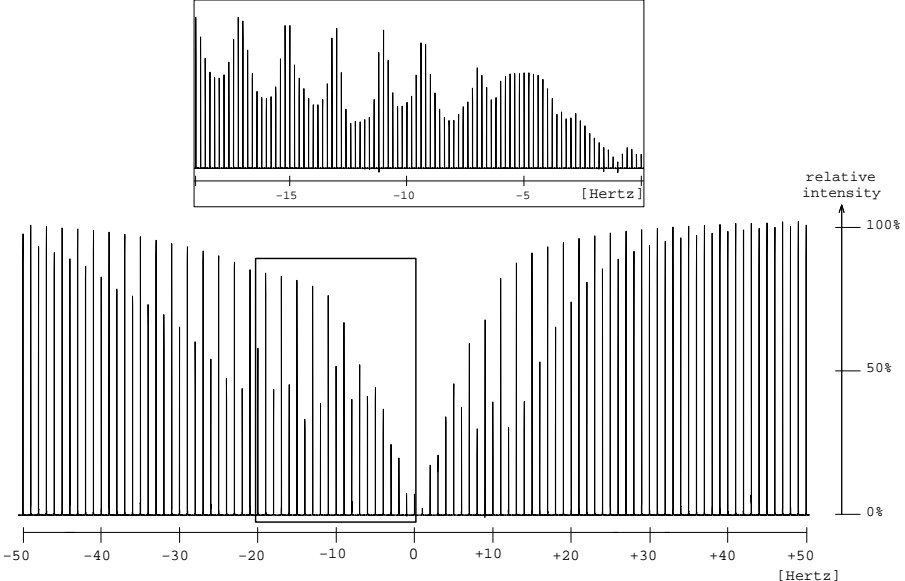

**Figure 7.** Presaturation profile in the region of two oxygen-bearing carbons for the 1,2–propanediol. A focus is made between 0 and -20 Hz to make the wavy part of the profile more visible.

## 4   Experimental

Glycerol, 1,2–propanediol, 1,3–propanediol, 1,2–butanediol and 1,3–butanediol solutions were prepared by addition of $200\,\mathrm{mg}$ of each to $0.6\,\mathrm{mL}$ of DMSO–$d_6$. This corresponds to sample with high concentration: $3.62\,\mathrm{M}$ for glycerol, $4.38\,\mathrm{M}$ for propane-diols, and $3.70\,\mathrm{M}$ for butanediols. Glycerol was kindly donated by Pierre Fabre Dermo-Cosmétique. 1,2–propanediol and D(+)–

sucrose were purchased from VWR. 1,3–propanediol and 1,3–butanediol were purchased from Alfa Aesar. 1,2–butanediol, was purchased from Sigma Aldrich. The sample containing D(+)–sucrose ($29\,\mathrm{mM}$) and glycerol in $0.6\,\mathrm{mL}$ of DMSO–$d_6$ was left overnight at room temperature to obtain a homogeneous solution.

All experiments were performed at $298\,\mathrm{K}$ on a Bruker Avance AVIII-600 spectrometer (Karlsruhe, Germany) equipped with a cryoprobe optimized for $^1$H detection and with cooled $^1$H, $^{13}$C, and $^2$H coils and preamplifiers. $^{13}$C NMR spectra

were acquired at $150.91\,\mathrm{MHz}$, with a $36\,\mathrm{kHz}$ spectral width and $32\,\mathrm{K}$ complex data points recording resulting in a $0.91\,\mathrm{s}$ FID acquisition time. The pulse length for excitation was $13.7\,\mathrm{\mu s}$ and the relaxation delay was $3\,\mathrm{s}$. Spectra were referenced for a central signal of DMSO–$d_6$ at $\delta$ 39.52.

The computer source code used in the present study was written in C language; it relied on the libxml2 library for the reading of the input data file (this may be an overkill for such a task, admittedly) and on the libsimu1 library for the calculation

of rotation matrices by means of Eqn. (4), as programmed for the design of SLURP pulses. The libsimu1 archive file also contains a proof of Eqn. (4). The computer code for saturation simulations is available from GitHub; its installation was tested

with Cygwin in Windows 10 but should be easily carried out on any other platform that provides a C language compiler and UNIX–like tools.

## 5   Conclusions

The present work provides a method for the saturation of intense solvent resonances in $^{13}$C NMR spectroscopy, as those occurring during the analysis of complex plant extracts prepared in high boiling point solvents. The signal reduction of these solvents was successfully achieved using the multi–site presaturation technique.

Numerical simulation therefore helped us to understand the origin of unexpected presaturation profile related to the saturation of frequency close resonances, even though it neither takes into account instrumental shortcomings such as $\mathbf{B}_0$ and $\mathbf{B}_1$
field inhomogeneities nor incomplete relaxation between transient signal recordings. The evolution of the sample magnetization was determined through the use of a simple approximation for the resolution of the Bloch equations that might find applications in other contexts. This approach offers perspectives in signal suppression from other natural sample matrices and in the quantitative $^{13}$C NMR analysis of extracts diluted in high boiling point solvents.

*Code and data availability.*   The PresatSimul source code is available from https://github.com/nuzillard/PresatSimul. The libsimu1 source
code is available from https://github.com/nuzillard/Libsimu1. The data files, pulse sequence and script from which Figs. 3 and 5 were obtained and a supplementary Figure and caption can be downloaded from https://www.zenodo.org/record/3635970 and temporarily from https://mycore.core-cloud.net/index.php/s/kQHZs7GaUxaZmmr.

*Author contributions.*   MC: Sample preparation, recording of NMR spectra, data processing and analysis, manuscript writing (in part). JMN: Supervision of the project, writing of the computer code for numerical simulation, of the initial code for automated data acquisition, and of
the manuscript (in part). SP, RR, and JHR: Manuscript text and figure reviewing. JHR and JMN supervised the PhD thesis work of M. Canton related to methodology developments in plant extract fractionation and NMR. All authors read and approved the final manuscript.

*Competing interests.*   The authors declare that they have no conflict of interest.

*Acknowledgements.*   M. Canton thanks the Laboratoires Pierre Fabre Dermocosmétique for financial support. Financial support by CNRS,
Conseil Régional Champagne Ardenne, Conseil Général de la Marne, Ministry of Higher Education and Research (MESR) and EU-
programme FEDER to the PlAneT CPER project is gratefully acknowledged. The authors thank the Association Nationale de la Recherche et de la Technologie for the financial support of the CIFRE grant #2017/1032. A. Martinez and A. Robert from ICMR, are also gratefully acknowledged for their technical support.





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
