# Peer review of "Multiple solvent signal presaturation in 13C NMR"

_Magnetic Resonance, 2020_

## Referee Comment (RC1) · Anonymous Referee #1 · 28 Apr 2020

This article entitled: "Multiple solvent signal presaturation in 13C NMR" by Canton et al presents a multi-site presaturation technique for 13C NMR spectroscopy. I have found the results and the approach nice and appropriate. I, therefore, think it is suitable for publication.

Nevertheless, I have a few minor comments and criticisms that authors could consider for the revised version of their manuscript.

1) the degree of novelty of the method was barely discussed. I think it is beneficial to have a brief statement summarising the originality of the approach so that it can stand out among other attempts.

2) At first, it was not clear to me at all - and still, it is not entirely - that what is the main purpose of the proposed method: solvent- or artefact suppression? i.e. it is not clear that the suppression of heterodecoupling artefacts is the goal of this study or a rather

attractive by-product of it? This point needs to be clarified. Especially if the artefact suppression is the primary goal, then the title itself is a bit misleading and probably should bear an indication of artefact suppression for that matter.

3) Figure 3 is somewhat confusing; there is no noticeable intensity difference between solvent signals in (a) and (b). The only difference between (a) and (b) that I can see is the disappearance of heterodecoupling artefacts, which is very nice. Still, I expected to see some intensity difference with respect to solvent signals themselves as well.

4) Some other attempts with regard to multiple solvent suppression are worth mentioning, including (Teodor Parella, 1998) and (Claudio Dalvit, 1998).

———————————————————

---

## Referee Comment (RC2) · Anonymous Referee #2 · 11 May 2020

The manuscript would be of interest to many people. The following suggestions may improve an already good paper:

Replace 13C with 13C(1H}. This is important because the authors report problems when the decoupler.

The proton concentration is not 110 M, is around 99 M as biological samples contain around 10% of deuterated water.

It would be nice if the authors comment on the use of decoupler schemes that are tolerant to pulse imperfections.

The authors should clarify whether the problems with the decoupler not being calibrated appear even when the probe is well tuned. Do they have an auto-tuning probe? In some cases, the miscalibration problem can be minimized with auto-tuning probes. In

other cases, pulse calibrations are still necessary.

Replace "Miscalibrations may cause" with "Decoupler miscalibrations may cause". Miscalibrations on the carbon channel rarely cause problems.

―――――――――――――――

---

## Author Comment (AC1) · 17 May 2020

The authors thank the referees for their comments and suggestions.

**Referee #1**

1. *the degree of novelty of the method was barely discussed. I think it is beneficial to have a brief statement summarising the originality of the approach so that it can stand out among other attempts.*

   Surprisingly, a bibliographic search about signal suppression in $^{13}$C NMR did not lead to adequate bibliographic references in this field. This point has been made clear in the revised manuscript.

2. *At first, it was not clear to me at all - and still, it is not entirely - that what is the*

*main purpose of the proposed method: solvent- or artefact suppression? i.e.
it is not clear that the suppression of heterodecoupling artefacts is the goal of
this study or a rather attractive by-product of it? This point needs to be clarified.
Especially if the artefact suppression is the primary goal, then the title itself is a
bit misleading and probably should bear an indication of artefact suppression for
that matter.*

The initial idea was to remove the observed artefacts. Then came the idea they
might be due to decoupling, and that intensity reduction of strong signals would
also bring their artefacts much below the noise level. The title has been changed
accordingly to "Multiple solvent signal presaturation and decoupling artifact re-
moval in $^{13}$C{$^1$H} NMR", a change that also takes into account a suggestion from
Referee #2.

A new sentence indicates "The advantage drawn from this operation (saturation)
is not only the intensity reduction of the solvent signals but also the elimination of
the possible artifacts that arise from the solvent signals in non–optimized decou-
pling conditions".

3. *Figure 3 is somewhat confusing; there is no noticeable intensity difference be-
tween solvent signals in (a) and (b). The only difference between (a) and (b) that
I can see is the disappearance of heterodecoupling artefacts, which is very nice.
Still, I expected to see some intensity difference with respect to solvent signals
themselves as well.*

Figure 3 has been redrawn and its caption changed accordingly to show the
solvent intensity reduction by presaturation.

4. *Some other attempts with regard to multiple solvent suppression are worth men-
tioning, including (Teodor Parella, 1998) and (Claudio Dalvit, 1998).*

The suggested bibliographic references to works by Parella and Dalvitt dealing
with multiple presaturation have been added.

**Referee #2**

- *Replace 13C with 13C{1H}. This is important because the authors report problems when the decoupler.*

  "$^{13}C\{^1H\}$" has been inserted in the title and the text.

- *The proton concentration is not 110 M, is around 99 M as biological samples contain around 10% of deuterated water.*

  The text states now that the concentration of protons in biological samples is close to 100 M.

- *It would be nice if the authors comment on the use of decoupler schemes that are tolerant to pulse imperfections.*

  No other decoupling scheme has been investigated as now stated in the text.

- *The authors should clarify whether the problems with the decoupler not being calibrated appear even when the probe is well tuned. Do they have an auto-tuning probe? In some cases, the miscalibration problem can be minimized with auto-tuning probes. In other cases, pulse calibrations are still necessary.*

  The probe is always auto–tuned. However, even with optimal tuning and matching, the actual length of the 90 degree pulse (whatever the RF channel) may vary depending on the nature of the sample. Moreover, automating tuning and matching in automation mode is not always optimal. A sentence was added to make this point clearer.

- *Replace "Miscalibrations may cause" with "Decoupler miscalibrations may cause". Miscalibrations on the carbon channel rarely cause problems.*

  The sentence that starts with "Miscalibrations may cause" has been modified according to referee's proposal.

---

## Author Comment (AC2) · 18 May 2020

Please note that a new title has been proposed for manuscript mr-2020-6: Multiple solvent signal presaturation and decoupling artifact removal in 13C{1H} NMR.
* * *